

**Study on monitoring and numerical analyses of**
**groundwater variation and inclinometer displacement**
**induced by heavy typhoon rainfall**
**Ching-Jiang JENG[1], Chia-Yu Yang [2]**
[1]{ Associate Professor, Huafan University, New Taipei City, Taiwan }
[2]{ Project Assistant, Huafan University, New Taipei City, Taiwan }
Correspondence to: Ching-Jiang JENG (jcjhf@cc.hfu.edu.tw)
**Abstract**
This study examined the slope area of the Huafan University campus, within the entire Ta-
Lun Shan area. Situated at 550 m above sea level, the area is geographically classified as a dip
slope. For sustainable development of the campus, a variety of continuous monitoring
equipment has been set. This study continues to analyze the monitoring data in the previous
period, but focuses on the observation and comparison of the horizontal displacement at
different depths recorded by inclinometers with time. At the same time, the relationship
between the groundwater variation caused by heavy typhoon rainfall and the driven deep
displacement of the slope is studied. In addition, this study has also been extended to the
whole Ta-Lun Shan area, the observation data of cracks inside and outside the campus are
analyzed. The displacement and sliding data collected over the years by the inclinometers are
used to study the potential sliding surfaces distribution of the slope, combined with the
influence of the typhoon rainfall. This study also continues to obtain the relationship between
groundwater level and rainfall, such as the time lag and groundwater level changes, and
reviews the effectiveness of the implementation of two catchpits at the campus. In the process
of renovation, a Shape Acceleration Array (SAA) is used for measuring the instantaneous
displacement to monitor if the displacement has increased during the implementation of the
catchpits. The SAA measurement is taken as the reference basis for hazard prevention and
maintenance of the slope, thus reducing and preventing the occurrence of disasters in the area.



Then, a series of shear tests are carried out. Finally, the software Geostudio is applied to
analyze the relationship between rainfall infiltration, groundwater level, and the displacement
of the entire slope. The numerical analysis results are compared with the monitoring results in
order to comprehensively discuss the slope safety.
**1   INTRODUCTION**
This research focuses on the dip slope area at the Huafan University in northeastern Taiwan.
The campus is located on a geological dip slope toward the southwest in the Ta-Lun Shan area
with an elevation ranging from 450 to 550 m, as shown in Fig. 1. For risk management and
research on slope stability, a monitoring system was set up in 2001, which has collected
geographical data since then. The monitoring system includes inclinometers, tiltmeters, crack
gages, water-level observation wells, settlement and displacement monitoring marks, rebar
strain gages, concrete strain gages, and rain gages.
Jeng and Sue (2016) analyzed the monitoring data collected from more than 300 settlement
and displacement observation marks on this site and compared them with the displacement
recorded by the inclinometers, finding a preliminary relationship between the displacement of
the slope and daily rainfall on the campus. The program STABL based on the limit
equilibrium method was applied for analysis, concluding that a rise in the groundwater table
caused by typhoons is the most critical factor in slope stability. Therefore, several
countermeasures, including catchpits with horizontal drainage pipes, were recommended, and
threshold-value curves of the slope displacement based on rainfall intensity and cumulative
rainfall were established. Those curves were derived from the rainfall records of numerous
typhoon events over the past ten years, along with the corresponding slope displacement
increment recorded by the ground surface marks.
The Japan Association for Slope Disaster Management (JASDiM) recommended the threshold
values of slope displacement for different sliding stages, which were used to define three
ranges. However, the displacement data recorded by the inclinometers used in the study were
retrieved once a month, and the displacement and subsidence of the slope by the observation
points were observed every six months. Therefore, the instantaneous changes of each typhoon
rainstorm event could not be known. An additional groundwater level gauge was installed in
the study area, and two Shape Acceleration Arrays (SAA) were installed within the



inclinometer casings, which have observed depths of sliding surfaces. This has made it
possible to obtain the continuous changes of groundwater level and slope displacement during
typhoon rainfall. Slope stability was also analyzed during the period.
Considering the project budget, two catchpits were implemented within the study area in the
present stage. The study provides valuable monitoring information and experience as
continuation of campus observations for years. The study also offers a comprehensive survey
of the slope behavior during rainfall infiltration, including the highest groundwater level
measured within the slope, the time lag reaction of groundwater variations, the influence of
rainfall types and amount, the relationship curve of rainfall amount and groundwater level,
and a review of the effectiveness of the two catchpits. The displacement curve measured by
the inclinometers was used for feedback comparison, and numerical analysis and simulation
were performed to validate the changes in the behavior and mechanism. Then, according to
the characteristics of the slope, including the allowable displacement alert and action values,
the data analyses revealed a corresponding relationship rainfall curve. The results will help
assess possible changes in groundwater level within the slope, possible slope displacement,
and safety and stability factors from the estimated rainfall before a disaster occurs.
Generally, slope displacement can be distinguished into several stages, in which the three
stages are: "initial displacement," "constant velocity displacement," and "accelerated
increment displacement." Xu (2011) pointed out obvious stages characteristics for the gradual
evolution of slope variations. To classify a type of displacement into one of the three stages,
the s-t curve for the relationship between displacement and time can be converted into a T-t
curve. The curve after conversion has a unique and deterministic tangent angle ($\alpha$). Normally,
the tangent angle of the curve is greater than 45 degrees when it enters the acceleration stage.
Detailed demonstration of each deformation stage is shown in Figs. 1(a) and (b).
**2   SITE INFORMATION**
The Huafan University is located on a hillslope. Over the years, with the support of the
university and the Ministry of Science and Technology, more than 40 inclinometers (Fig. 2),
two surface extensometers, 48 rebar drain gauges, 36 concrete strain gauges, and 28 tiltmeters
have been installed at the campus. In addition, more than 300 ground subsidence and
displacement observation points with ground surface and structural crack detection records.
All of these monitoring instruments are continuously operational in taking regular
measurements. This combination of a wide range of experimental site information is a



valuable repository of data for both national and international research. With the joint efforts
of various authors in the past years, this site has focused on the monitoring system, geological
and hydrological data, and the establishment of a geographic information system, generating a
significant amount of research results (see Chao, 2002, 2003; Jeng, 2003, 2007; Jeng and Sue,
2009, 2016; Jeng and Lin, 2011, 2014; Jeng and Jiang, 2013; Jeng et al., 2015, 2017).
The main exposed stratum under the site is the Mushan Formation, with the bedrock being
mainly interbedded with sandstone and shale. It is a dip slope striking toward the east, dipping
southward about 10°–20°. Huang and Jeng (2004) pointed out that through the observation
and comparison of drilling cores and ground resistance images, as well as topographical
characteristics, there are two local small faults: a south-trending Nanshihkeng Fault and a
northwest-trending A Fault, as shown in Figure 3.
In addition to the above monitoring system, surface crack observation within the site is
conducted every six months. Cumulative observation results show continuous growth of the
number and size of the surface cracks over time. Therefore, continuous renovation and
reinforcement works have been carried out for the past few years. Figure 4 is the location map
of the surface cracks inside and outside the campus.

## 3   STUDY RESULTS AND DISCUSSIONS

**3.1 Analyses of monitoring results of auto-recording groundwater level gauges**

According to the basic information of the research site and the inclinometers, tiltmeters,
groundwater level gauges, building inclination measurements, displacement and subsidence
observation points, surface crack investigation and other monitoring equipment system data,
as well as the geological and hydrological data, analyses of the automatic groundwater level
monitoring results at the campus are as follows.
Analyses of historical data show that rainfall causes the slope displacement. In order to
understand the relationship between rainfall and the groundwater variation and the slope
stability, four sets of auto-recording groundwater-level gauges are designed in this research
area for observation and discussion. The configuration diagram of the groundwater -level
gauge (piezometer) is shown in Fig. 2. Additionally, Fig. 5 through Fig. 8 illustrate some of
the monthly rainfall data and groundwater-level changes from 2013 to 2016.
Figure 5 is a diagram of rainfall data and groundwater-level duration curves in April 2013.
The x-axis is the time, the left y-axis is the groundwater level (expressed as depth below GL),
and the right y-axis is the rainfall. The groundwater level in W1 has a dark blue curve, the

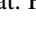



groundwater level in W2 has a red curve, the groundwater level in W3 has a green curve, and
the groundwater level in W4 has a purple curve. The light blue curve is the rainfall in April.
Figure 5 indicates that three heavy rainfalls occurred in April, on 4/1–4/5, 4/8–4/13, and
4/17–4/20. Their rainfall types can be classified as post-peak rainfall, pre-peak rainfall, and
pre-peak rainfall, respectively. Additionally, when rainfall occurs, groundwater level also
changes, although with some time lag. The degree of groundwater level change is
W1>W2>W3>W4, which indicates that degree of change of the groundwater level in
observation wells is different across different regions.
The monitoring data from Fig. 6 to Fig. 8 of the rainy and typhoon seasons in 2015–2016 was
analyzed using the same method. Similarly, the W1 groundwater level change is particularly
evident, with the groundwater level change uplifted by rainfall up to about 23 m, while in W2,
W3, and W4, the amount of change was about 8 m, 6 m, and 1 m, respectively. This indicates
that the groundwater level in the area of W1 was more susceptible to rainfall influence. If
heavy rainfall happens in the future, the area near W1 must be closely monitored. A
comparison of different rainfalls shows that the peak rainfall of post-peak rainfall was greater
than the peak of other types of rainfall, so post-peak rainfall should be monitored more
closely. Furthermore, in Fig. 5, it is shown that during the rainy season in April, the rainfall
period is short, and the rainfall amount is mostly less than 40 mm; the groundwater level rise
speed is slow, and the rising amount is small. Figures 6–8 report the rainfall data from the
typhoon season, from August to October, during which the rainfall is mostly greater than 40
mm. Additionally, during the typhoon season, rainfall will cause greater rise of the
groundwater level, and the lag time between the peak rainfall and peak groundwater level will
significantly shorten. When the groundwater level in W1 rises to about ground level (GL) -30
m, W2 and W3 will then increase more noticeably. Based on the above information, during
the typhoon season, W1 can be used as an important observation index for alerts and
protective measures to avoid problems with the slope stability change.
Given the above analyses, it is clear that groundwater-level change caused by rainfall can vary
due to (1) rainfall type, (2) rainfall amount, (3) rainfall time lag, (4) groundwater-level gauge
region, and (5) seasonal factors. Analyses of these factors are illustrated in Figs. 9–11, which
contain only analysis data from observation well W1. These figures are, in this order, the
groundwater level and rainfall peak time lag diagram, the peak rainfall and groundwater-level
rise diagram, and the accumulated rainfall and groundwater-level rise diagram for W1



groundwater-level gauge from 2013 to 2015.
Figure 9 is the groundwater level and rainfall peak time lag diagram of automatic
groundwater-level gauge W1. The x-axis is the peak rainfall, and the y-axis is the time lag for
groundwater level to reach its peak. Figure 10 is the peak rainfall and groundwater level rise
diagram of automatic groundwater-level gauge W1. The x-axis is the peak rainfall, and the y-
axis is the amount of groundwater level rise. Figure 11 is the accumulated rainfall and
groundwater-level rise diagram of automatic groundwater-level gauge W1. The x-axis is the
accumulated rainfall, and the y-axis is the groundwater-level rise amount. The round dots are
post-peak rainfalls, the square dots are pre-peak rainfalls, and the triangle dots are middle-
peak rainfalls. There are two types of accumulated rainfall: accumulated up to the peak (dark-
colored symbols) and accumulated for the entire rainfall (light-colored symbols).
Figure 9 shows that the greater the peak rainfall, the shorter the reaction time lag is for the
groundwater level to rise. With respect to the rainfall type perspective, the peak rainfall of the
pre-peak rainfalls is concentrated less than 20 mm and has a long groundwater-level time lag.
The peak rainfall of the middle-peak rainfalls ranges from 20 mm to 40 mm, and the peak
rainfall of the post-peak rainfalls is concentrated at around 50 mm with a short time lag. It
indicates that if this study area has rainfall greater than 50 mm, it should be closely observed
and the observation time should be stretched to about 40 hours. Further, Fig. 10 shows that for
greater peak rainfall, the groundwater level will significantly increase, and the post-peak
rainfall will cause the most significant groundwater level rise. According to the auto-
recording groundwater-level gauge, the groundwater rise amount was between 1 and 23 m.
Lastly, Fig. 11 shows that when the total average cumulative rainfall to peak point exceeds
200 mm, the groundwater level in auto-recording groundwater-level gauge in W1 will
increase up to 20–23 m.
W2–W4 were observed and analyzed by using the same method, and the results indicate that
if the rainfall type is the post-peak, the time lag for its groundwater level to reach the peak is
the shortest, resulting in the largest increase in its groundwater level. Moreover, when average
cumulative rainfall exceeds 200 m, significant groundwater-level increase will occur,
requiring close observation for up to 40 hours.
As mentioned above, when the peak rainfall is greater, the time lag of the groundwater level
rise will be shorter. The peak rainfall of the post-peak rainfall type is greater than that of other
rainfall types, which makes the change of its groundwater level significant. Thus, observation


for up to 40 hours is needed for the post-peak rainfall. The data from the auto-recording
groundwater-level gauge shows that the groundwater level W1 is most affected by rainfall,
with a change between 1 m and 23 m. A groundwater-level change of 0.6–5.8 m was found in
the auto-recording groundwater-level gauge in W2, 0.25–3.1 m in W3, and 0.1–2 m in W4.
**3.2 Analysis and discussion of inclinometer displacement**
The location of the inclinometers at the site is shown in Fig. 2. This study used
cumulative rainfall amount from previous typhoon alerts and the displacement of 29
inclinometers to observe the displacement of the inclinometer casings and aggregated the data
into graphs to show the trend. The data are organized sequentially based on the gradual
evolution of slope change over time as recorded in previous research. The present study only
used nine casings with a relatively large displacement and only the analysis of inclinometer
SIS-32 was used as an example, as shown in Fig. 12:
Figure 12 is the SIS-32 displacement and cumulative rainfall amount from typhoon rainfall.
The x-axis is the cumulative rainfall amount from typhoon rainfall, and the y-axis is the
displacement caused. Different inclinometer displacements from different typhoons in the
past years are presented in the graph. The data curve pattern was classified as gradual change
type referring to the damage curves pattern classification presented by Xu (2011) shown in
Fig. 1. The graduated pattern means that the displacement will be different (for different
rainfall amount) at different times, and the change will accelerate when a certain rainfall
amount is reached. Once the displacement starts to accelerate, the accumulated rainfall is
treated as an alert rainfall threshold value. Figure 12 shows the typhoon cumulative rainfall
alert of SIS-32 has a threshold of about 350 mm.
The typhoon cumulative rainfall alert threshold values from the other nine inclinometers are
summarized in Table 1, and positions of the inclinometers with accelerated displacement
history are plotted on the map in Fig. 13. Table 1 shows that the alert cumulative rainfall
threshold for heavy typhoon rainfall causing campus slope displacement is approximately
315–495 mm. The threshold is relatively low in the Sport ground and near Wu-Ming Building,
meaning that it has a higher degree of hazardous urgency. Figure 13 shows that the available
data thus far show that regions at the campus where heavy typhoon rainfalls reached alert
threshold value are: (1) the area surrounded by Wu-Ming Building, Hui-Tsui Building, and
the Library Building; (2) from the Sport ground to the slope of Material laboratory; and (3)
from Lotus garden outside the campus to Ta-Lun road. Therefore, these three regions should

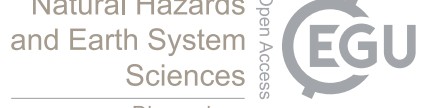



be the focus of slope safety maintenance and future monitoring of slope displacement during
typhoon rainfall.
The campus already has existing groundwater monitoring data, as previously discussed, to
understand the lifting and possible flow of the groundwater at slope above the A Fault.
However, the campus slope has experienced large displacement during the rainy season when
the groundwater level is high. Considering the above data analyses, it is recommended that
the Wu-Ming Building at the campus should be accorded priority consideration in future
heavy rainfall events for safety improvement needs. Accordingly, two catchpits were
implemented below and above the Wu-Ming Building slope (Fig. 2) to reduce or adjust the
groundwater level for enhanced slope safety and stability.
**3.3 Maximum groundwater level monitoring data and analysis to explore the**
**effectiveness of catchpits**
In this study, simple groundwater level gauges were mounted to manually obtain the highest
groundwater-level readings for observations and discussions. The six highest groundwater-
level gauges were mounted to inclinometers SIS-1, SIS-9, SIS-11, SIS-26, SIS-32, and SIS-38,
as shown in Fig. 2. The highest groundwater-level-variation data from some of the
inclinometers from April to July 2016 are plotted in Figs. 14–15.
Figure 14 is the relation graph of rainfall data and groundwater level from the auto-recording
groundwater-level gauge from April to July 2016. The x-axis is time, the left y-axis is the
water level, and the right y-axis is the rainfall amount of the day. The groundwater level from
the auto-recording groundwater-level gauge in W1 has a red curve, the water level in W2 has
a green curve, the water level in W3 has a purple curve, and the groundwater level in W4 has
a light blue curve. The dark-blue curve shows the daily rainfall amount from April to July.
The figure shows that, similar to aforementioned data, W1 is much more affected by the
rainfall amount than the other three auto-recording groundwater-level gauges. The data from
the auto-recording groundwater-level gauge in W1 from 2013 to 2016, as shown in Fig. 15,
shows that the normal groundwater level from the auto-recording groundwater-level gauge in
W1 has a lowest value of about −43.67 m from 2013 to 2015. After the implementation of the
catchpits in 2016, the lowest curve has decreased significantly to −52.03 m. The highest
groundwater level also dropped from −22.67 m to −27.84 m. This is due to the
implementation of the two catchpits at the site, which significantly reduced both normal
groundwater level and the highest groundwater level during rainy and typhoon seasons.



Figure 16 is the groundwater level rise amount distribution from the aforementioned six
highest groundwater-level gauges. It is clear from the figure that the groundwater level
variation in the region between the A Fault and the Nanshihkeng Fault is very large (mostly
higher than 21m as shown in Fig. 16), while the groundwater-level variation in the region
below the Nanshihkeng Fault is relatively small. The current findings thus conclude that SIS-
32 located out of the area enclosed by the Nanshihkeng Fault and the A Fault, enabling the
groundwater from rainfall infiltration to smoothly flow down the slope, so the groundwater-
level rise (6-8m) is less than the region surrounded by the two faults.
**3.4 Discussion of SAA displacement measurement results**
The current manual measurement of the inclinometers occurs at the frequency of once a
month. The drawback is that it is unable to measure the correct amount of displacement
caused by heavy typhoon rainfalls and that there are errors caused by manual measurements.
In order to improve the situation, the SAAs have been implemented since 2014 at 3 m long
above and below the sliding depth at the inclinometers SIS-11 and SIS-20 in the vicinity of
Wu-Ming Building (Fig. 13). The SAA sensors are implemented every 0.5 m, with total 12
sensors setting in each inclinometer. Starting from 2017, two new SAAs are added for SIS-1
and SIS-8. In addition to the benefits of its continuity (one measurement every 10 min), high
precision, wide range of deformation measurement and automatic readings to compensate for
the drawbacks of manual reading of inclinometers, another purpose for the use of SAA is to
monitor if excessive slope displacement occurs before and after the implementation of
catchpits, in order to better fulfill the purpose of the warning system.
This study used rainfall data and SAA displacement from April 2014 to November 2017 to
observe and analyze the relationship between them. Data were aggregated to show trends,
such as in Fig. 17, which contain data captured and amplified from June 2014 as examples.
Figure 17 is respectively the relationship between inclinometer displacement of SIS-11 and
SIS-20 and rainfall amount from June 2014. The x-axis is the date, the left y-axis is the daily
rainfall amount, and the right y-axis is the daily accumulated displacement of inclinometers
SIS-11 and SIS-20. On June 19 and 23, 2014, there were heavy rainfall (hourly rainfall
amount > 100 mm), and the displacement also rapidly increased due to two continuous heavy
rainfall, demonstrating that displacement increases with rainfall amount, and continuous
heavy rainfall will make the slope displacement even more evident.
The above analyses show that the displacement varies due to the following factors: (1) rainfall





amount, (2) continuous heavy rainfall, and (3) environmental conditions at the inclinometer
position. The cumulative rainfall and SAA displacement increment information is
summarized in Table 2, which shows the obtained cumulative rainfall amount and the
increment of the cumulated displacement amount.
Using the data in Table 2, the minimum driven displacement of daily rainfall turns out to be
77 mm/day; if it is less than 77 mm/day then it belongs to steady state. The other point worth
noting is that the data obtained from Table 2 are much less than the comprehensive results
over the years (Jeng and Sue, 2016). This is because since the implementation of SAA until
November 2014, no heavy typhoon rainfalls took place during monitoring. Thus, Table 2 is a
general reference for ordinary heavy rainfall. The analysis below concerns situations relevant
to typhoon rainfall.
This study sought to understand slope displacement caused by typhoon rainfall, so the
collection of rainfall data and SAA displacement data continued until February 2017 to
observe and analyze the relationship between typhoon rainfall and displacement of SAA. The
results are aggregated into charts to observe trends, as shown in Fig. 18 and Fig. 19.
Figures 18 and 19 are respectively the relationship graph between the displacement of
inclinometer SIS-11 and inclinometer SIS-20 and the rainfall amount. The x-axis is the month,
the left y-axis is the rainfall amount, and the right y-axis shows the accumulated displacement
of inclinometer SIS-11 and inclinometer SIS-20. These figures show a significant increasing
trend of displacement during July–September 2015 and September–October 2016, and the
displacement of SIS-11 is slightly larger than the displacement of SIS-20. In this paper, only
part of the data was taken and enlarged, shown in Figs. 20–21. The x-axis is the date, the left
y-axis is the rainfall amount, and the right y-axis shows the accumulated displacement of
inclinometer SIS-11 and inclinometer SIS-20.
As shown Figures 20 to 21, there were two typhoons in August 2015, the medium typhoon
Soudelor and the strong typhoon Goni. Accordingly, SIS-11 had displacement of 6.048 mm
and 1.489 mm, respectively, and the displacement of SIS-20 was 4.839 mm and 1.64 mm,
respectively. In October 2016, there was light typhoon Aere along with its peripheral
circulation. SIS-11 had a consequent displacement of 11.321 mm, while SIS-20 had a
displacement of 3.384 mm. The above data are plotted in Fig. 22 as an SAA displacement
increment graph during typhoon rainfall.
Figure 22 shows that when the typhoon cumulative rainfall reaches 184 mm, significant




displacement will occur; when the cumulative rainfall reaches 300 mm or more, or when the
peak rainfall exceeds 156 mm, the displacement will begin to accelerate, producing
displacement larger than 3 mm. This result agrees with that in Section 3.2 of this study.
Therefore, the study sets typhoon rainfall of more than 300 mm as the alert threshold. In other
words, if the weather forecast reports a typhoon rainfall of more than 300 mm, then the
displacement situation should be closely monitored.
When considering the ordinary typhoon situation, the daily cumulative rainfall and SAA
displacement increment relation diagram can be extended to February 2017. The result shows
that a daily cumulative rainfall of up to 300 mm the displacement will accelerate. In a typical
typhoon situation, the average formula can be used to deduce the displacement increments.
The results are plotted in Table 3 for future reference.
**3.5 Discussion of Geostudio simulation results**
In this study, the aforementioned geological and hydrological data of the site were entered
into GeoStudio software to carry out SEEP/W, SLOPE/W, and SIGMA/W numerical analysis.
The simulation results were compared to the monitoring data to analyze the effect of rainfall
on the slope.
**3.5.1 SEEP/W analytical results**
This study analyzed the initial condition setting: boundary conditions of rainfall infiltration
are shown in Fig. 23, wherein AB is a boundary of the lower constant head slope, and the
total head height is set to H = 387.8 m; CD is the head boundary of the upper slope, and the
total head height is set to H = 534.5 m. The upper and lower boundaries of the head setting
mainly use the average groundwater level from historical monitoring data. The BC is the
impervious boundary (Q = 0); AD is the boundary of rainfall infiltration. Table 4 shows the
input parameter for the analysis, Fig. 24 is the soil groundwater characteristic curve, and Fig.
25 shows the input value of hydraulic conductivity curve, which is mainly based on the
resulting of colluvium soil test in pressure plate. These two parameters form a function of
groundwater characteristic curve (Fig. 24), and then the groundwater characteristic curve is
converted into hydraulic conductivity curve (Fig. 25).
This study used rainfall data during typhoons in August 2015 and October 2016, the amount
of groundwater-level variations from four auto-recording water-level gauges, as well as two
auto-recording inclinometer SAAs, to observe and analyze the relationship between rainfall
amount during typhoon, slope displacement, and groundwater-level variation. The results are





aggregated to show the trend. The groundwater level data and the automatic inclinometer
SAA data were taken from the Huafan University slope center. The rainfall data were taken
from the weather station at the Huafan University.
Figures 20 and 21 show that two typhoons struck in August 2015, the typhoon Soudelor
during 8/6–8/9, and the typhoon Goni during 8/20–8/23. Soudelor generated a rainfall amount
of 342.5 mm, an SIS-11A displacement increment of 6.048 mm, and an SIS-20A
displacement increment of 4.893 mm. Typhoon Goni generated a rainfall amount of 30.5 mm,
an SIS-11A displacement increment of 1.489 mm, and an SIS-20A displacement increment of
1.64 mm. A southern low-pressure system (typhoon Aere) generated a rainfall amount of 794
mm, an SIS-11A displacement increment of 11.321 mm, and an SIS-20A displacement
increment of 3.384 mm. The rainfall amount and actual displacement data were used as a
reference in the GeoStudio software simulation analysis.
The groundwater-level simulation results from Figs. 26–28 demonstrate that groundwater
level will increase during typhoon. The groundwater level variation is 0.6–1.65 m during the
typhoon Soudelor, 0.09–1.16 m during the typhoon Goni, and 0.01–5 m during the southern
low-pressure system (Aere typhoon). This indicates that different typhoon rainfall amount
caused different groundwater-level variations: the greater the typhoon rainfall amount is, the
greater the groundwater-level varies, but the degree of groundwater level variations is smaller
than the actual monitoring data (average 18 m). The possible factors are (1) the hydraulic
conductivity of the simulated soil, (2) infiltration reduction coefficient, (3) the water head of
slope, (4) element mesh size, (5) normal rainfall amount in steady state, (6) actual position of
cross-sectional soil model, and (7) boundary conditions of cross-sectional variations and the
parameter settings. Since most of the parameters of this study were hypothetical, there is a
certain distance from the actual situation, and the results of this study can only be used for
initial determination, which should be supplemented by further research.
**3.5.2 SIGMA/W analytical results**
SIGMA/W displacement analyzed its initial boundary condition settings, as shown in Fig. 29.
The boundaries AC and ED are the fixed boundaries of horizontal displacement; CD is set to
be the fixed boundary of vertical and horizontal displacement; AE applies building loads
according to the cross-sectional position of the buildings, as shown in Fig. 29. There are three
buildings in the cross-section profile: Hua-Fan Temple, Wu-Ming Building, and Jue-Zhao
Building. The loads applied are 10 kPa, 50 kPa, and 50 kPa, respectively. Table 5 shows the





settings of the remaining loads, and Table 6 shows the input parameters for the SIGMA/W
numerical model analysis.
Figure 30 shows the SIS-20 and SIS-11 inclinometer displacement increment diagram during
the typhoon Soudelor; Fig. 31, the typhoon Goni; and Fig. 32, the southern low-pressure
system (Aere typhoon). These figures demonstrate the displacement result simulated by this
model has the same displacement curve trend as the actual monitoring data. There is a slightly
different analytical value at the shear zone, with an alignment error value of only 0–2 mm. A
possible reason for the error is that simulated soil layer mesh has uneven nodes. In future
studies, this module can be modified to apply the continuous shear test to adjust the input
parameters and soil layer mesh size to change the settings. The actual displacement value
from the typhoon rainfall observation can be further expanded for comparative analysis, in
order to arrive at a more accurate prediction of the slope displacement results caused by future
typhoon rainfalls and to provide the campus a reference for slope safety.
Based on the above results, it is presumed that the reason of simulation results showing
difference in the present observation data is that the actual displacement amount of SAA is all
within 20 mm (2 cm). For cumulative rainfall and peak rainfall smaller than 60 mm, the
change in the output result by the simulated rainfall is extremely small, therefore leading to
the error. The result agrees with the results mentioned above in Table 2; generally, only when
the rainfall exceeds 77 mm will the displacement be driven. In other words, simulation results
from the software using cumulative rainfall and peak rainfall greater than 60 mm will be more
important and more meaningful than simulation results from simulating a single-peak rainfall.
Future studies should conduct the simulation in this manner.
**3.5.3 SLOPE/W analytical results**
The strata strength parameter input for the analysis of SLOPE/W was the same as the input
parameter value of SIGMA/W model analysis.
The analysis method adopted two modes:
(1)The minimum safety factor of sliding surface was calculated within the specified range of
the slope. (2) Since the current inclinometer monitoring records clearly show the sliding
surface depth, the sliding surface known is adopted in the numerical model for slope stability
analysis, and the sliding depth is connected from each inclinometer record as a reference for
determining the sliding range. This study adopted the Morgenstern-Price extreme balance
method to calculate the safety factor of the sliding surface. Figure 33 shows the profile of





SLOPE/W analysis. The most critical potential sliding surface existed between Jue-zhao building and Wu-ming building is shown in the profile. The results of safety factor variation during typhoon are shown in Fig. 34.

Fig. 34 shows that during the typhoons Soudelor and Goni, although the safety factor of the slope will drop to about 0.4, the minimum safety factor for rainfall can still be maintained at 1.3. This illustrates that even though typhoon rainfall will cause a certain amount of slope displacement at the campus, its safety factor is still higher than the general suggested value of 1.2; in other words, there is not yet immediate danger of sliding. However, further studies should investigate the impact on safety factors in situations of higher typhoon rainfall through actual follow-up test results and feedback analyses.

## 4    CONCLUSIONS AND SUGGESTIONS

This paper offers a comprehensive survey of the slope behavior during rainfall infiltration, including the highest groundwater level measured within the slope, the time lag reaction of groundwater variations, the influence of rainfall types and amount, the relationship curve of rainfall amount and groundwater level, and a review of the effectiveness of the two catchpits. The displacement curve measured by the inclinometers is used for feedback comparison, and numerical analyses and simulations are performed to validate the changes in the behavior and mechanisms.  Based on the results, the following conclusions can be made:

1. When the peak rainfall is higher, there is less delay of groundwater level rise, and the peak rainfall of post-peak rainfall is larger than that of other rain types, which makes the groundwater-level variation large. Thus, observation up to 40 hours is needed for post-peak rainfall type. The groundwater-level W1 is most affected by the rainfall, with a variation of 1–23 m. The groundwater level variation was 0.6–5.8 m for W2, 0.25–3.1 m for W3, and 0.1–2 m for W4.

2. After the implementation of the catchpits, the lowest groundwater level has decreased from −43.67 m in 2015 to −52.03 m in 2016. The highest groundwater level also dropped from −22.67 m to −27.84 m. Therefore, initial observations confirmed that the normal groundwater level and highest groundwater level has been significantly decreased during the rainy seasons and typhoon seasons by the draining effectiveness of the two catchpits.



3. The cumulative rainfall amount and SAA displacement increment both exhibit a certain variation trend, and obtained average formula and the data in Table 2 can be used as a reference for ordinary heavy rainfall. In a typical typhoon situation, the average formula can be used to deduce displacement increment, as shown in Table 3, for future reference.

4. When the typhoon cumulative rainfall reaches 184 mm, significant displacement will occur; when the cumulative rainfall reaches 300 mm or more, or when the peak hourly rainfall exceeds 100 mm, the displacement will begin to accelerate, producing displacement of larger than 3 mm. Therefore, the study sets typhoon rainfall of more than 300 mm as the alert threshold. If the weather forecast reports a typhoon rainfall of more than 300 mm, then the displacement situation should be closely monitored.

5. The Geostudio groundwater-level simulation results show that the groundwater level will increase during typhoons. The groundwater level variation was 0.6–1.65 m during the typhoon Soudelor , 0.09–1.16 m during the typhoon Goni, and 0.01–5 m during the southern low-pressure system (typhoon Aere). However, the degree of groundwater-level variations was smaller than the actual monitoring data (average is 18 m), possibly due to differences in the hydraulic conductivity of the simulated soil, infiltration reduction coefficient, element mesh size, boundary conditions of cross-sectional variations, and parameter settings. Further studies are necessary for refined model correction.

6. The typhoon Soudelor simulation shows that the displacement result simulated by this model has the same displacement curve trend as the actual monitoring data. There is a slightly different analytical value at the shear zone, with an alignment error value of only 0–2 mm. A possible reason for the error is that simulated soil layer mesh has uneven nodes. In future studies, this model can be modified to apply the continuous shear test and adjust the soil layer mesh size to change the settings. The actual displacement value from typhoon rainfall observation can be further expanded for comparative analysis, in order to reach a more accurate prediction of slope displacement caused by future typhoon rainfalls and to provide the campus a reference for slope safety.

7. The displacement amount SAA is all within 20 mm. For cumulative rainfall and peak rainfall smaller than 60 mm, the change in the output result by the simulated rainfall is extremely small, therefore leading to the error. The result agrees with the results mentioned above; generally, only when the rainfall exceeds 77 mm will the displacement be driven. In other words, simulation results from the software using cumulative rainfall and peak rainfall




greater than 60 mm will be more important and more meaningful than simulation results from
simulating a single peak rainfall. Future studies should conduct the simulation in this manner.
8. The slope model analytical results show that during the typhoons Soudelor and Goni,
although the safety factor of the slope will drop to 0.4, the minimum safety factor for rainfall
was still maintained at 1.3, higher than the general suggested value of 1.2. This means that
there is not yet any immediate danger of sliding. However, future studies should follow up
with the actual test results and feedback analysis to reach a more comprehensive
understanding of the impact of rainfall on the safety factors in situations of higher typhoon
rainfall.

**Author Disclosure Statement**

No competing financial interests exist.
**Acknowledgements**
This study is funded by the Ministry of Science in Taiwan as under project number MOST
105-2632-M-211-001. The authors are grateful for the support, which made it possible for the
study to proceed and conclude smoothly.

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



2    Table 1: Typhoon cumulative rainfall alert threshold values for each inclinometer

| Inclinometer number | Threshold (mm) |
|---|---|
| SIS-1 | 420 |
| SIS-5 | 420 |
| SIS-7 | 350 |
| SIS-17 | 420 |
| SIS-20B | 370 |
| SIS-25A | 420 |
| SIS-32 | 350 |
| SIS-33B | 315 |
| SIS-40 | 460 |

4    Table 2: Daily cumulative rainfall from heavy rainfall days and SAA displacement increment

| Term | Daily cumulative rainfall (mm / day) | SAA displacement increment (mm) |
|---|---|---|
| 1 | 77–150 | 0.21–0.34 |
| 2 | 150–300 | 0.34–1.03 |
| 3 | 300–500 | 1.03–1.95 |
| 4 | > 500 | 1.95 |





Table 3: Daily cumulative rainfall and SAA displacement increment for general typhoon
conditions.

| Term | Daily cumulative rainfall (mm / day) | SAA displacement increment (mm) |
|---|---|---|
| 1 | 68.5–150 | 0.42–0.56 |
| 2 | 150–300 | 0.56–0.80 |
| 3 | 300–500 | 0.80–4.17 |
| 4 | > 500 | > 4.17 |

Table 4: Input parameters

| Soil layer | Seepage mode | k (cm / s) |
|---|---|---|
| Colluviums layer | Saturation | $10^{-3}$ |
| Sandstone and shale interbedded | Saturation | $10^{-4}$ |
| Sandstone layer | Saturation | $10^{-5}$ |
| Shear zone | Saturation | $1.39 * 10^{-5}$ |

Table 5: Building load parameters

| Building name | Applied load (kPa) |
|---|---|
| Hua-fan temple | 10 |
| Wu-ming building | 50 |
| Chea-chau building | 50 |





1    Table 6: Input Parameters for numerical analysis

| Soil layer | Colluvium soil layer | Sandstone and shale interbedded | Gravel layer | Shear zone |
|---|---|---|---|---|
| Deformation model | Elastoplastic | Elastoplastic | Linear elasticity | Interface elements |
| Elastic modulus (kPa) | $3 \times 10^4$ | $3.225 \times 10^5$ | $3.753 \times 10^6$ | 3748 |
| Poisson's ratio | 0.334 | 0.28 | 0.23 | 0.334 |
| Unit weight (kN/ m$^{2)}$ | 19.31 | 25.52 | 23.86 | 23.3 |
| Cohesion, C (kN/ m$^{2)}$ | 18.5 | 41.8 | 38.7 | 0 |
| Internal friction angle, $\Phi$ (degrees) | 29.6 | 32.13 | 32.74 | 23 |



**Figures and captions**

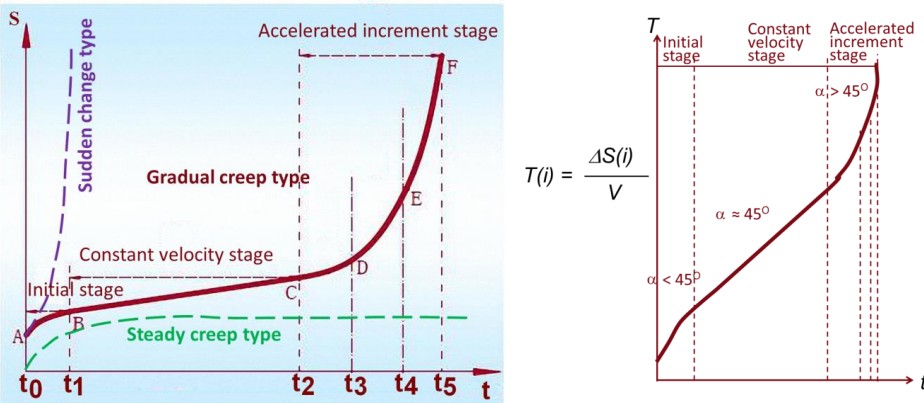

Figure 1: S-t curve of slope sliding displacement (a) and T-t curve of slope sliding
displacement (b) (modified after Xu et al., 2008 and Xu, 2011).

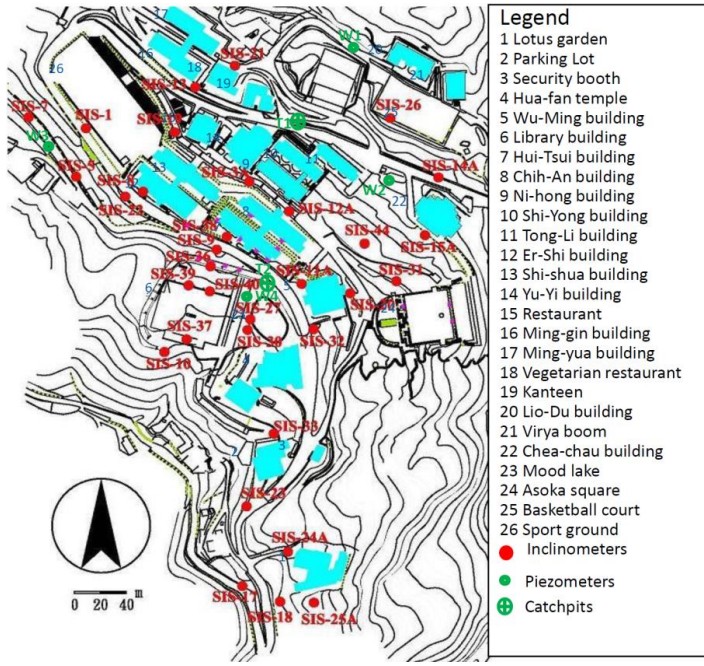

Figure 2: Inclinometer distribution at the Huafan campus





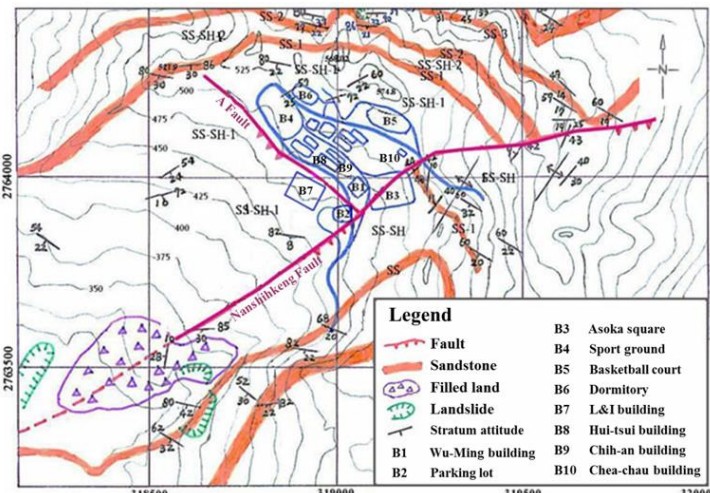

2  Figure 3: Geological map of the research area (Huang and Jeng, 2004)

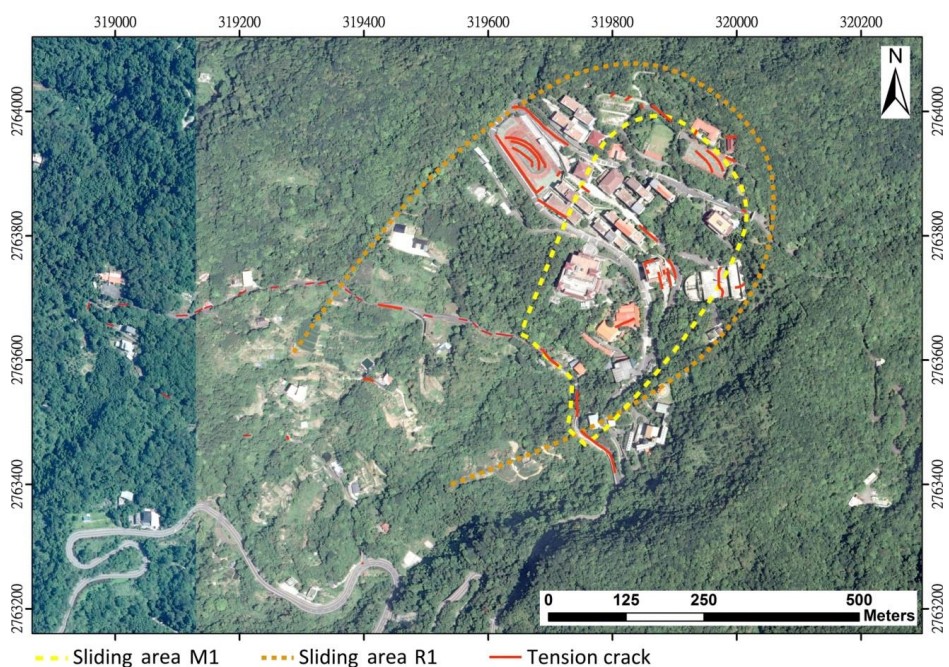

5  Figure 4: Location of surface cracks inside and outside the campus.



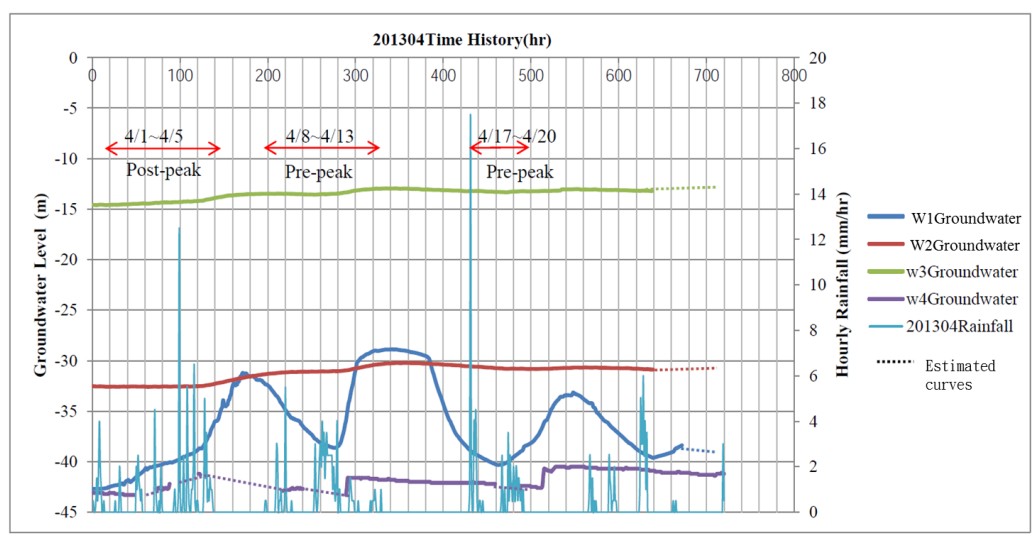

Figure 5: Groundwater level and rainfall duration curves from auto-recording groundwater-
level gauge in April 2013.
Figure 6: Groundwater level and rainfall duration curves from auto-recording groundwater-
level gauge in August 2015.



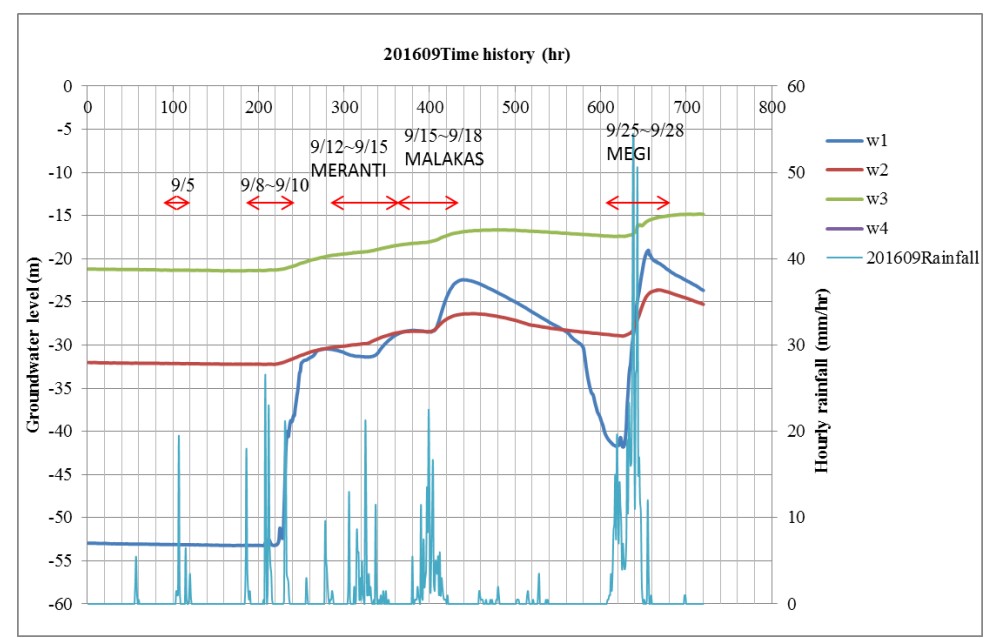

2      Figure 7: Groundwater level and rainfall duration curves from auto-recording groundwater-

3      level gauge in September 2016.



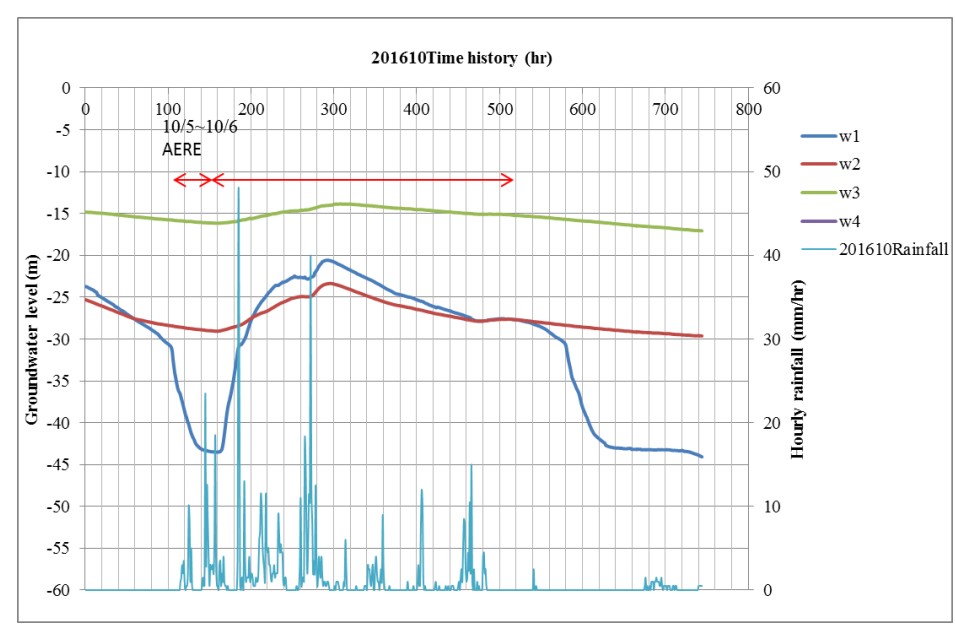

Figure 8: Groundwater level and rainfall duration curves from auto-recording groundwater-
level gauge in October 2016.

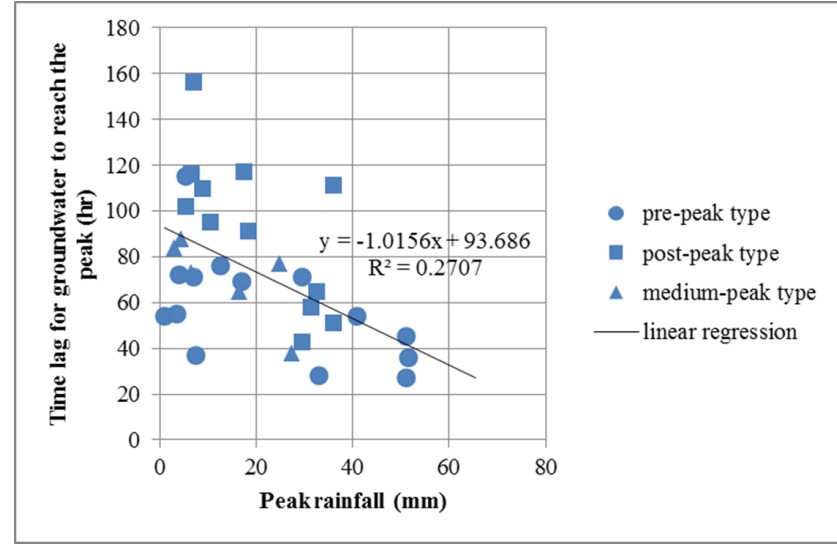

Figure 9: Groundwater level and time lag of rainfall peak diagram of automatic groundwater-
level gauge W1.



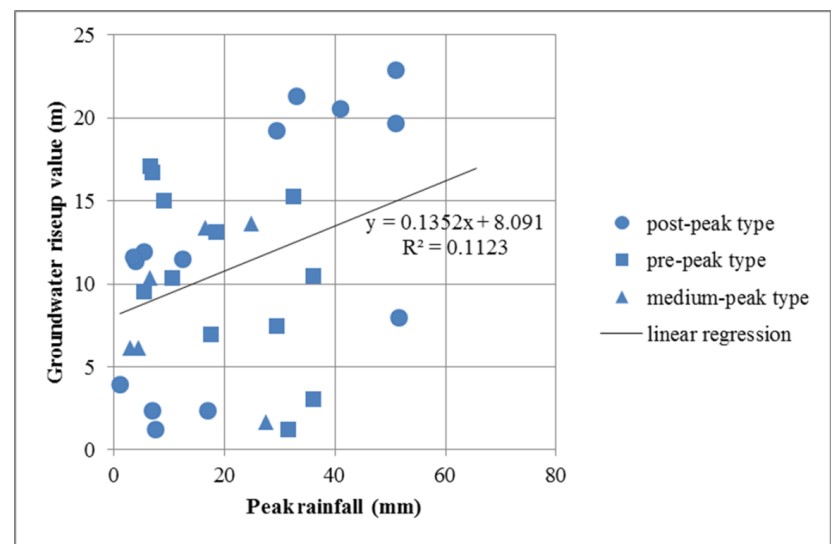

3    Figure 10: Peak rainfall and groundwater level rise diagram of automatic groundwater-level

4    gauge W1.

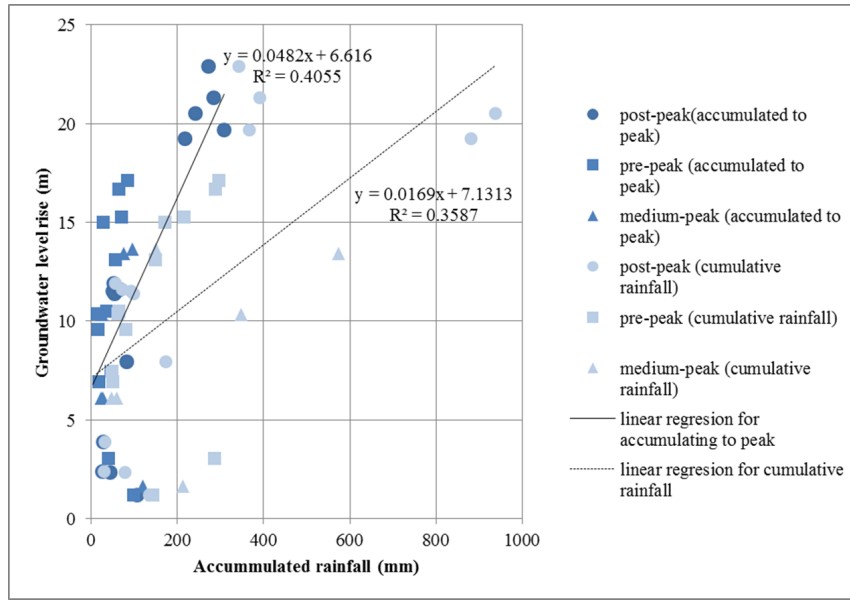

7    Figure 11: Accumulated rainfall and groundwater level rise diagram of automatic





1    groundwater-level gauge W1.

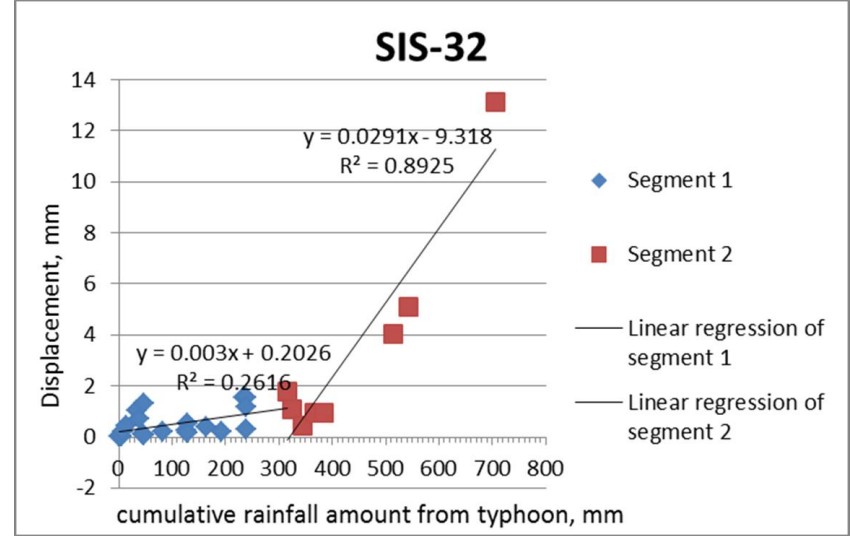

4    Figure 12: SIS-32 displacement and cumulative rainfall amount from typhoon rainfall.



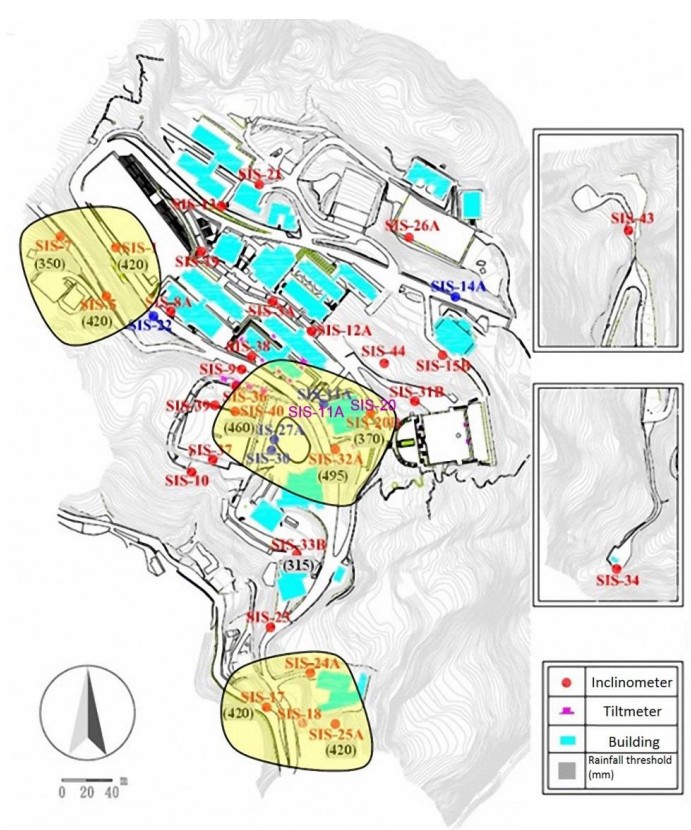

2    Figure 13: Positions of the inclinometers where alert rainfall threshold has been reached

3    during typhoon rainfalls.



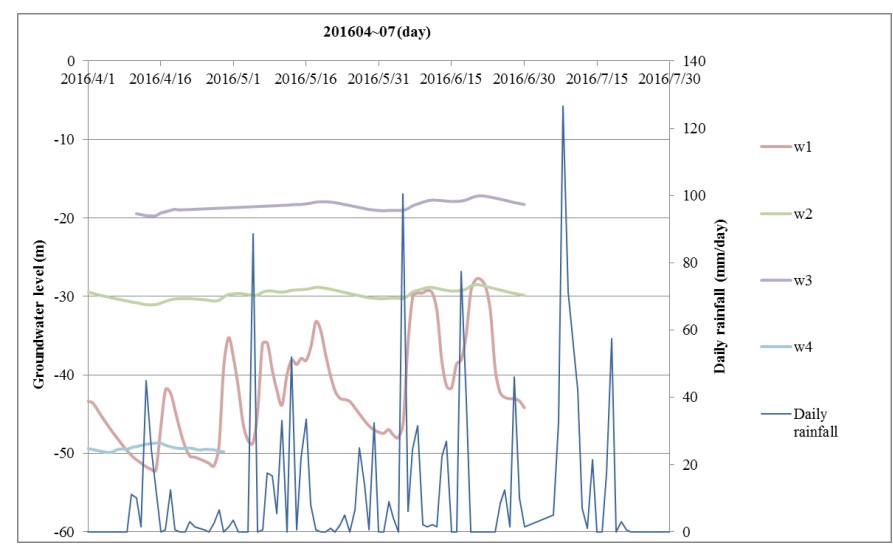

2    Figure 14: Time graph of groundwater level and rainfall amount from auto-recording

3    groundwater-level gauge from April to July 2016.

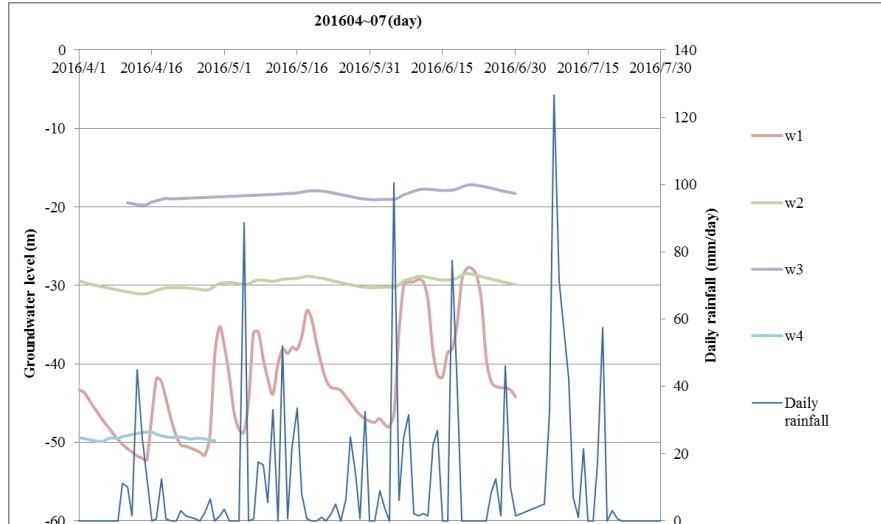

6    Figure 15: Comparison of groundwater level before and after catchpits implementation.




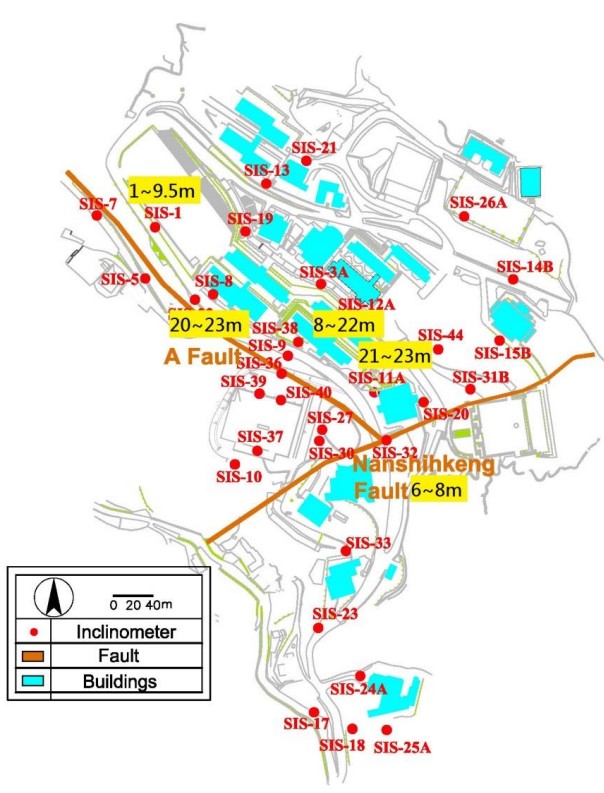

Figure 16: Groundwater level rise amount distribution from highest groundwater-level gauges
and SAA locations.
Figure 17: Relationship between SIS-11 and SIS-20 displacement and rainfall amount from
June 2014.





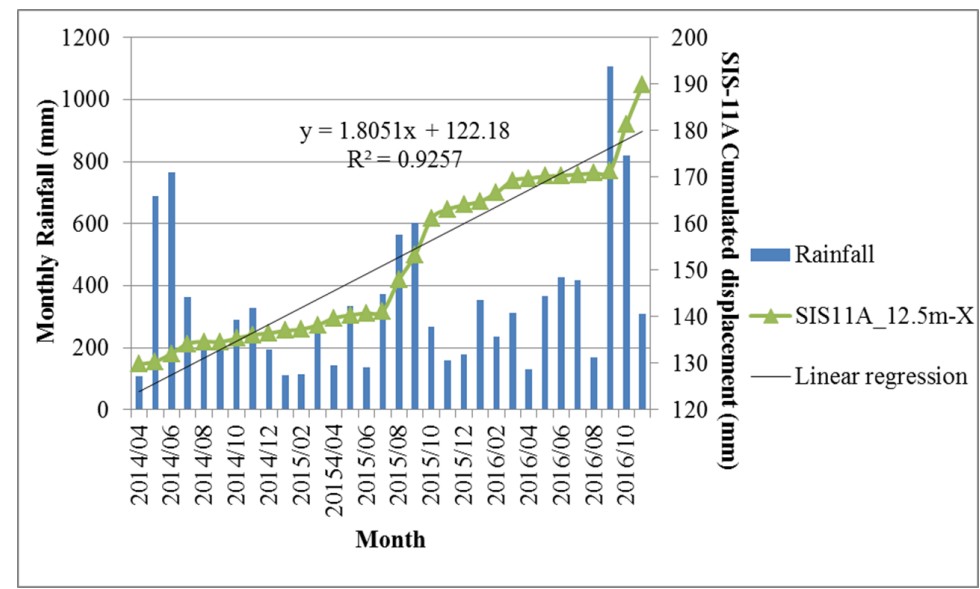

2    Figure 18: SIS-11 displacement versus rainfall amount (as in February 2017).

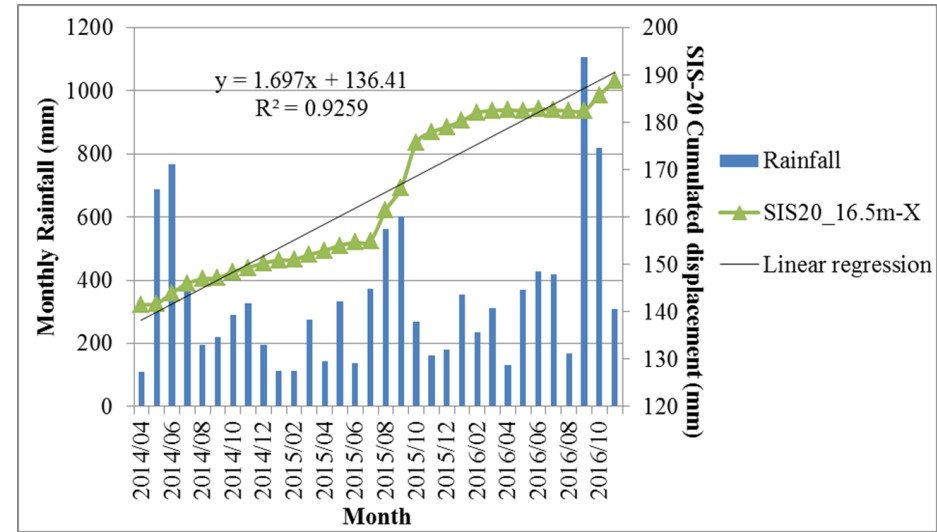

5    Figure 19: SIS-20 displacement versus rainfall amount (as in February 2017).





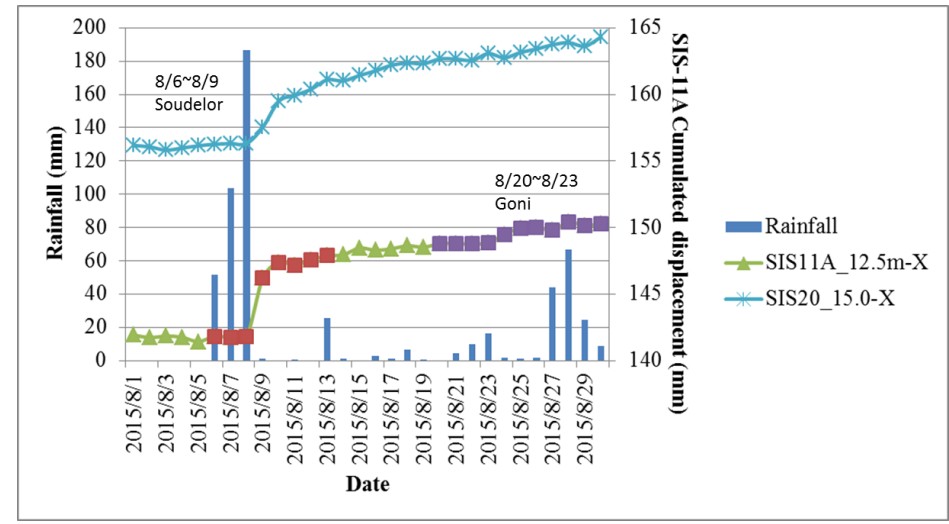

Figure 20: SIS-11 and SIS-20 displacement versus rainfall amount in August 2015.

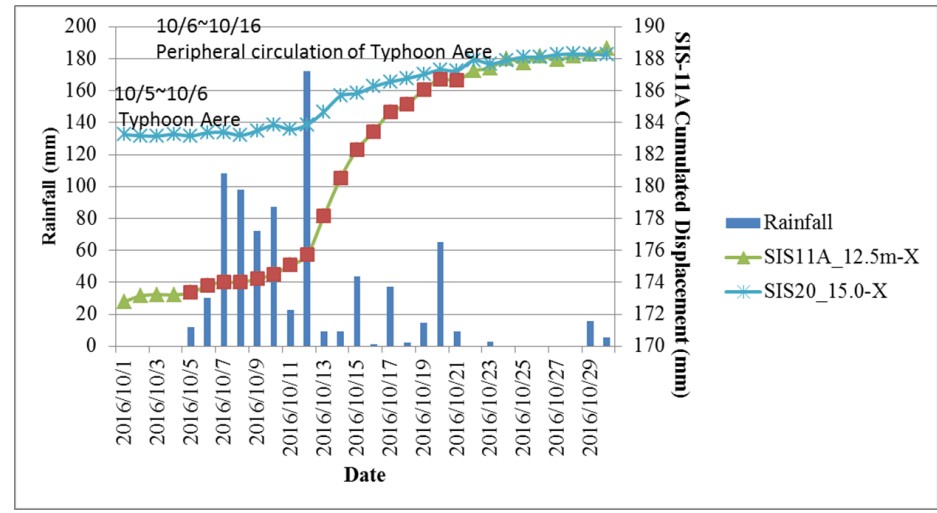

Figure 21: SIS-11 and SIS-20 displacement versus rainfall amount in October 2016.



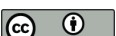

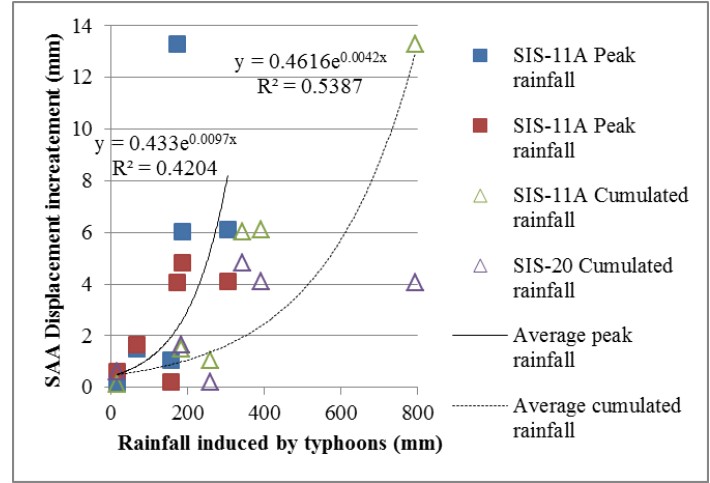

2    Figure 22: SAA displacement increment during typhoon rainfall.

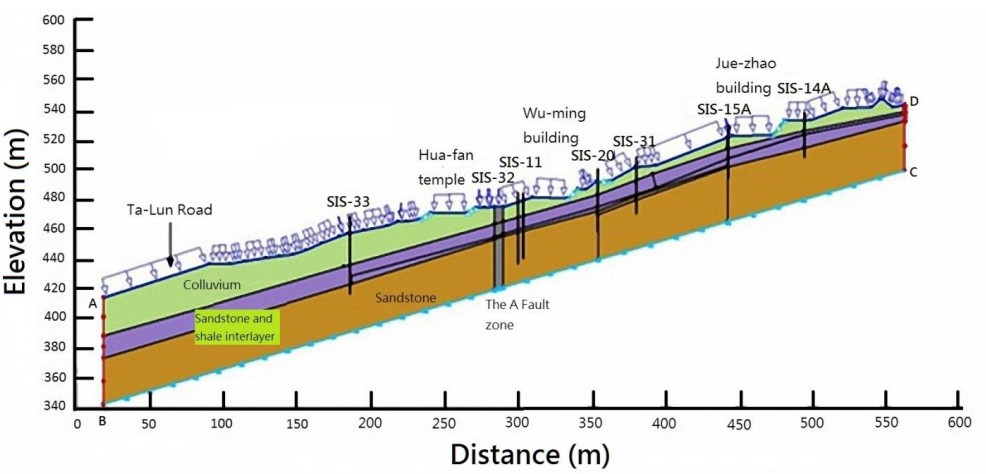

5    Figure 23: Cross-section profile of SEEP/W.



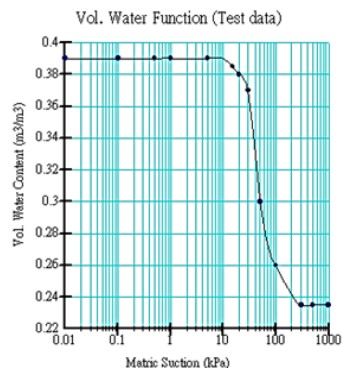

2      Figure 24: Soil water characteristic curve.

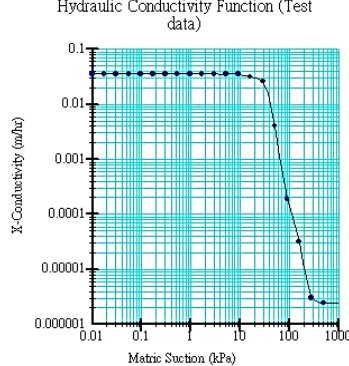

5      Figure 25: Hydraulic conductivity curve.



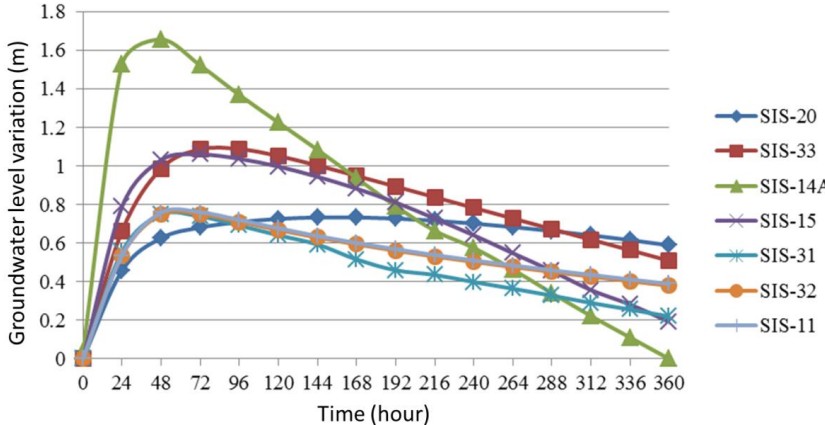

2    Figure 26: Simulation of groundwater-level variation during the typhoon Soudelor.

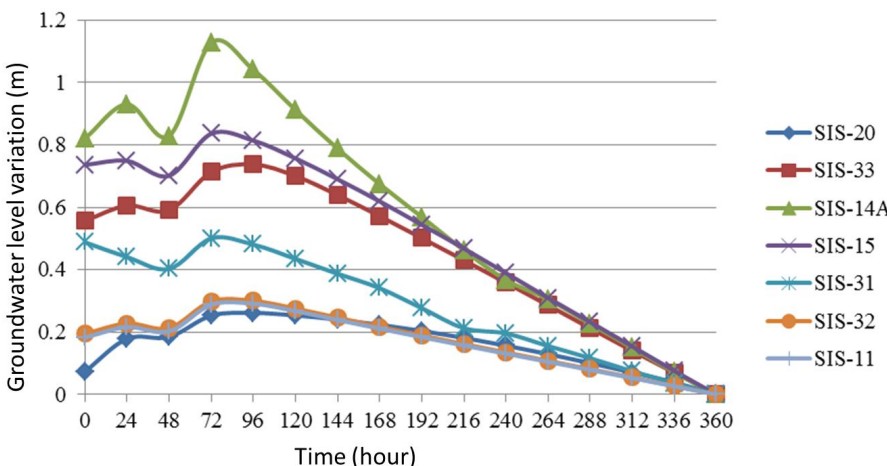

5    Figure 27: Simulation of groundwater-level variation during the typhoon Goni.





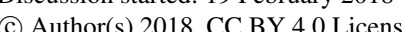

2    Figure 28: Simulation of groundwater-level variation for the southern low-pressure system

3    (Aere typhoon)

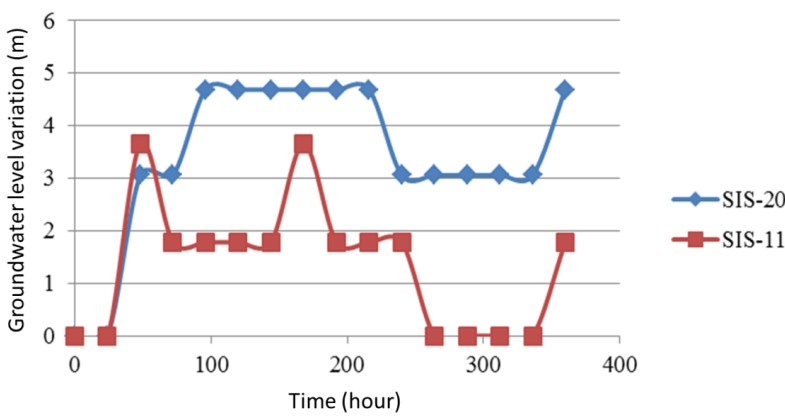

6    Figure 29: Cross-sectional view of SIGMA/W.





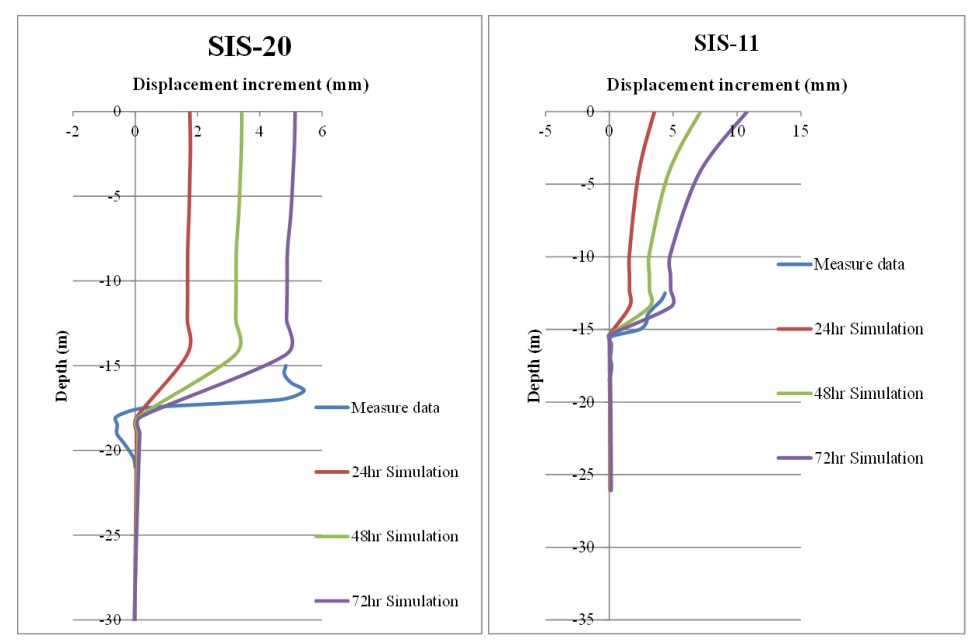

Figure 30: SIS-20 displacement increment (left) SIS-11 displacement increment (right) of
simulation of the typhoon Soudelor.

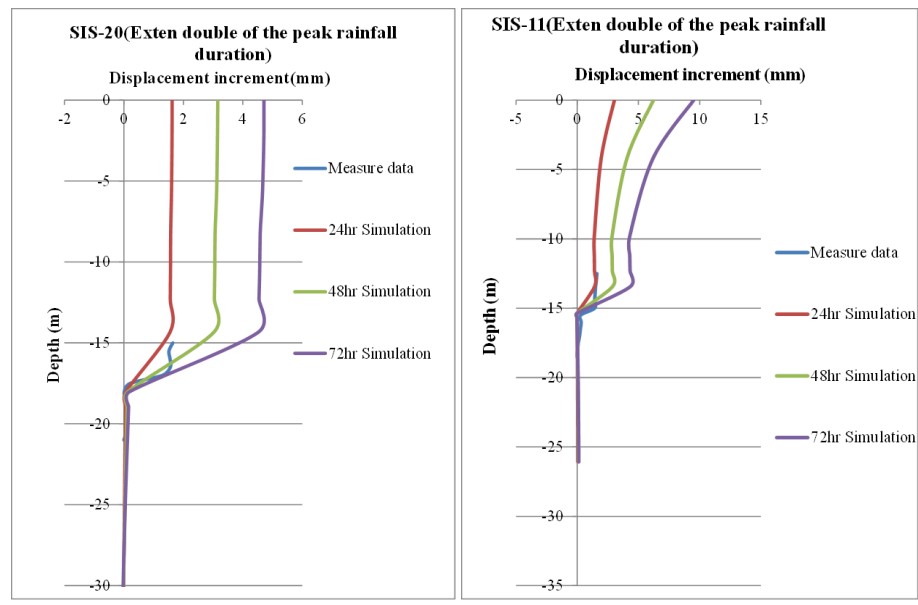

Figure 31: SIS-20 displacement increment (left) SIS-11 displacement increment (right)of
simulation of the typhoon Goni.



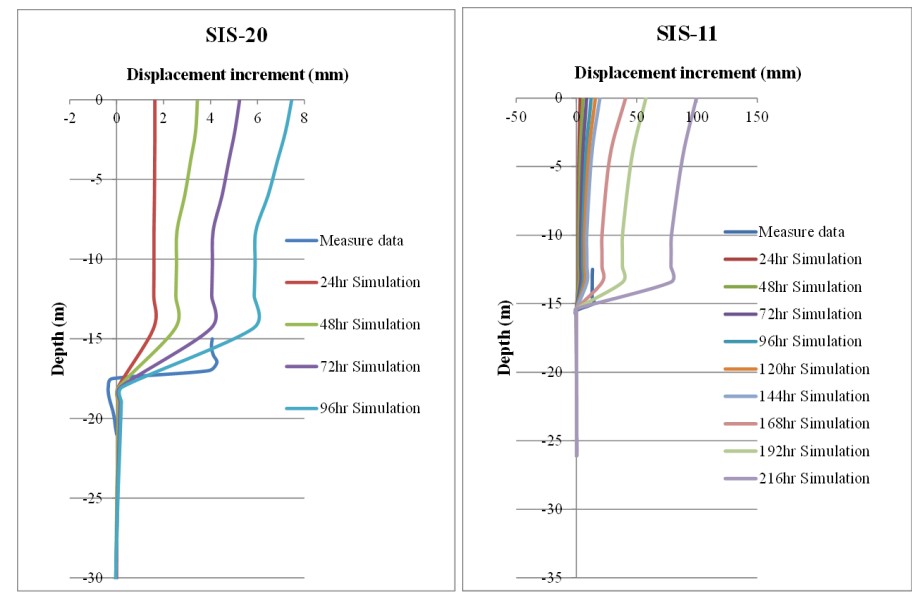

2 Figure 32: SIS-20 displacement increment (left) SIS-11 displacement increment (right) of

3 simulation of the southern low-pressure system (typhoon Aere).

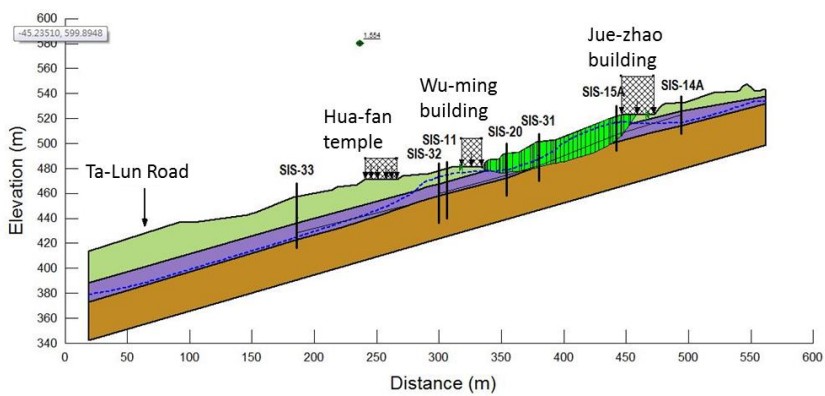

6 Figure 33: Profile of potential sliding surface analysis





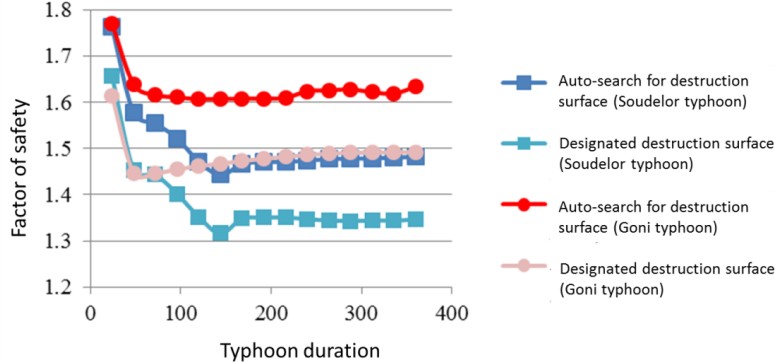

2      Figure 34: Variation of safety factor in typhoon simulation.