# Peer review of "Study on monitoring and numerical analyses of"

_Natural Hazards and Earth System Sciences, 2018_

## Short Comment (SC1) · 14 Apr 2018

This paper reveals many detailed data from the field measurements. It is interesting to be referred on slope engineering when interpreting the effects of heavy rainfalls on the ground water table variations and the stability as well as the deformations of the slope. Although many details of the observations have been discussed in this paper, it is suggested that the authors may enhance this article by adding up some suggestions out of their field observations. That is the remedy works which can increase the three dimensional global safety of the U. campus together with the factors of safety at

some simplified two dimensional cross-section profiles from the zones of the University campus could be elaborated. This would make this paper more sounding to the readers. The effectiveness of the current drainage system at the U. Campus and what should be installed for better improvements can be discussed in this paper based on the observations. Apart from that, it would be very interesting to carry out in the future studies the three dimensional FE analysis which can cover up the topographic and geological conditions of the site. This would be very interesting to see how good the remedy solutions will be and their impacts to the hazard preventions due to the rainfall and resultant ground-water changings.

---

## Author Comment (AC1) · 23 Apr 2018

We would like to thank Professor D.-W. Chang for your valuable comments and suggestions on this research. We will add up some suggestions based on the observations obtained from the study case in the revised manuscript. The effectiveness of the two drainage catchpits at the study site will be discussed more detailed as well in the revised paper. A three dimensional analysis has been carried out by using the discrete element method by the PFC 3D software, and the results have been presented in another published paper (as follow). Some description will be included and listed in the Reference chapter in the final manuscript. Reference: Chia-Han Tseng • Yu-Chang Chan • Ching-Jiang Jeng • Yu-Chung Hsieh, (2017) "Slip monitoring of a dip-slope and runout simulation by the discrete element method: a case study at the Huafan University campus in northern Taiwan," Nat Hazards 83(9) DOI 10.1007/s11069-017-3016-y, Springer. Published on line, 21 August, 2017.

---

## Referee Comment (RC1) · Anonymous Referee #1 · 9 Jul 2018

The authors present geotechnical data from the densely monitored campus area of the Huafan University in Taiwan. The campus is situated on a slowly displacing hillslope. The data consists of several displacement and hydrological observations. The authors also perform some basic correlation analysis and run commercially available software for pore water pressure and slope stability (GeoStudio software). The authors have presented most of the monitoring data before in several contributions among which in NHESS, 2016 (Jeng and Sue, Nat. Hazards Earth Syst. Sci., 16, 1309–1321, 2016).

Although the authors have an impressive data set, the manuscript is merely a geotechnical report: It give data, some numerical results (uncritically) but it lacks a scientific problem description and a clear objective. Scientifically, the authors do not present a novelty, not in data itself, nor in its analysis, not in the methodology and not in increased process understanding. And therefor, it is logical that the manuscript starts with "This research focuses on the dip slope area of the Huafan university." It is a description of a site, not of a scientific question. This becomes also evident when looking at the reference list: 14 references of which 10 (co-) authored by the principle author of this manuscript. In my opinion this is clear self-plagiarism and already for this reason the manuscript must be rejected in NHESS.

This technical report is in my opinion far from suited for NHESS-D or NHESS. I advice the authors to read national and international literature on hydrogeology and slope stability of deep-seated landslides, come up with intriguing and interesting research questions and use their fabulous data set to solve it. I am quite sure the authors should be able to produce interesting papers from this data set.

---

## Referee Comment (RC2) · Anonymous Referee #2 · 4 Dec 2018

This manuscript is aimed to find the relationship between typhoon rainfall, slope groundwater level and the displacement of the slope by monitoring data analysis and numerical analysis. This topic would be potentially interesting for the journal NHESS. However, the organization of the manuscript is poor that cannot matched the publication level right now.

1. Introduction The introduction part of this manuscript is like a report not in in an article style. It is not clear to show the research background and the highlights of this study.

3. Study results and discussions P6_12, "Figure 9 shows that the greater the peak rainfall, the shorter the reaction time lag is for the groundwater level to rise." Why? The correlation coefficient is small. The evidence is not enough to support this conclusion. P6_30, the question is the same as P6_12. P8_26, Fig.15 is the same as Fig.14 in P29, this should be a mistake. P10_30, The results in Fig.22 is not clear. Why combined data from two diffident monitoring points to analyze the threshold of the cumulative rainfall? The analysis for groundwater level is qualitative, here should be more quantitatively to show the effect of the rainfall to the groundwater level including the response lag time, response velocity and response rising degree. The analysis of the simulation work is also not clear and quantitatively to point out the relationship between typhoon rainfall, slop groundwater level and the displacement of the slope.

4. Conclusion The discussion of this manuscript is poor. The monitoring work of this study is worthy, but the scientific value of the conclusion is not enough. The manuscript is more like a report. It should be thoroughly revised.

---

## Author Comment (AC2) · 22 Jan 2019

We would like to thank the Referee for the valuable comments and suggestions on this research. This study collects the comprehensive monitoring data including rainfall, groundwater and slope displacement. However, partial data are derived from the case study, the classification of rainfall types, behavior of groundwater variation and displacement induced by heavy rainfall (during typhoons) are analyzed and discussed in detail. We believe that the mechanisms of slope displacement related to groundwater variations discussed in the study are valuable, and can be a reference for the study

on similar slopes. The authors have monitored this site and collected the information over more than ten years. Therefore, results through series of studies have been produced gradually over the period. The topics of each reference listed in this paper are different from the results of our research in each stage. Many of them were presented in Chinese language, and this may cause misunderstanding to the Referee. It should not be considered as self-plagiarism. Thanks for the Referee's comments. Further, specific literatures on hydrogeology and slope stability of deep-seated landslides will be added in the revised manuscript. The analysis results will be modified in the revised manuscript as well.

---

## Author Comment (AC3) · 22 Jan 2019

**Reply to Anonymous Referee #2:**

We would like to thank the Referee for the thoughtful reviews and comments.

The authors would like to thank that the Referee sympathizes with the topic of the manuscript. The organization of the manuscript will be modified and rearranged in the revised version.

1. Introduction: The contents of introduction have been rearranged and revised as shown in the appendix.

2. In Figure 9, the correlation coefficient is small because there are different types of rainfall i.e. pre-peak, post-peak and medium-peak, and each of them have different run-off effects. In addition, the data observed before and after the catchpits construction are put together, therefore the catchpits may have the effects of groundwater drawdown. After amendment, as shown in the revised figures 9(a) to 9(c), each rainfall type is distinguished respectively, and the data of before and after catchpits construction are illustrated in different symbols. The data indicate that for each rainfall type, before the catch pits were constructed, the correlation coefficient varies from 0.692 to 0.8226, and after the catch pits were constructed, the correlation coefficient varies from 0.643 to 0.5519. Furthermore, the time lag of response for the groundwater level to rise after the catchpits construction seems larger than that before the catchpits were constructed, especially for the pre-peak and medium-peak rainfall type (when the peak rainfall value is higher than 20 mm). That is because the catchpits draws down the groundwater and increases the time of groundwater level to rise. Further, by comparing the Figures 9(a), 9(b), and 9(c), it shows that the time lag value for the pre-peak type rainfall is observed higher compared to the other two types of rainfall. It is possibly due to the fact that in the pre-peak type of rainfall, most of the surface runoff from rainfall, before it seeps into the ground and raise the groundwater level. On the contrary, for the post-peak type of rainfall, before the rainfall amount reaches the peak value, the infiltrated rainfall may have gradually caused the upper soil to become wet or saturated, so that when the subsequent peak rainfall arrives, the groundwater level will easily rise. All the figures and discussions mentioned above will be added in the revised manuscript.

[Figure]

Figure 9(a): Groundwater level and time lag of rainfall peak diagram for pre-peak type rainfall.

[Figure]

Figure 9(b): Groundwater level and time lag of rainfall peak diagram for medium-peak type rainfall.

[Figure]

Figure 9(c): Groundwater level and time lag of rainfall peak diagram for post-peak type rainfall.

The results shown in Figure 10 and Figure 11 have been redrawn and discussed based on the catchpits construction effect, similar to the description for Figure 9, and shall be amended in the revised manuscript. The new figure shown as below is the relation curves of groundwater rise value vs. cumulated rainfall before and after the catchpits construction.

[Figure]

3. Fig. 15 is the same as Fig. 14, which is a mistake. The corrected Fig. 15 is shown as follows, and it will be also amended in the revised manuscript.

[Figure]

Figure 15: Comparison of groundwater level before and after catchpits implementation.

In the revised manuscript, the analyses of the simulation work will be revised for better clarity and to point out the quantitative relationship between typhoon rainfall, slop groundwater level and the displacement of the slope.

4. The conclusions will be revised based on the details mentioned above and amended in the revised manuscript.

APPENDIX:

**1   INTRODUCTION**

In general, slope displacement can be distinguished into various stages, in which the three particular stages are: "initial displacement," "constant velocity displacement," and "accelerated incremental displacement." Xu (2011) pointed out obvious characteristics of each stage for the gradual evolution of slope movement. To classify slope movement into the three stages, the s-t curve for the relationship between displacement and time can be converted into a T-t curve. The curve obtained following the conversion has a unique and deterministic tangent angle ($\alpha$). Normally, the tangent angle of the curve is greater than 45 degrees when it enters the acceleration stage. Detailed demonstration of each deformation stage is shown in Figure 1(a) and Figure 1(b).

Jesus et al. (2018) dealt with the constrains and triggering factors of landslides and mentioned that in addition to gravity, groundwater level is the most important factor in slope stability as it can affect slope stability in different ways, such as reducing the strength, changing the density and generating the pore water pressures. Furthermore, the study indicates that deep-seated landslides are also triggered by high groundwater levels; however; conditions for their occurrence are not so widely spread. Taib et al. (2017) implemented a sand slope model for testing its stability under rising groundwater condition. They found that the slope with the highest density has a higher safety factor, but results in faster slope failure due to rapid development of the pore pressure ratio. Prokešová et al., (2013) studied the monitored data, and their observations suggest that groundwater level (i.e. GWL) response to precipitation differs considerably with respect to both overall hydrological conditions and GWL mean depth. Further, they found that slightly above-the-average rainy season following the prolonged wet period can be far more responsible for the accelerated movement in deep-seated landslides, compared with the single season of extreme precipitation following a longer dry period.

The Japan Association for Slope Disaster Management (JASDiM) recommended the threshold values of slope displacement for different sliding stages, which were used to define three ranges. However, the displacement data recorded by the inclinometers used in the study were retrieved once a month, and the displacement and subsidence of the slope by the observation points on the ground were observed every six months. Therefore, the instantaneous changes of each typhoon rainstorm event could not be captured. An additional groundwater level gauge was installed in the study area, and two Shape Acceleration Arrays (SAA) were installed within the inclinometer casings, which were used to observe depths of sliding surfaces. These monitoring systems have enabled to obtain the continuous changes of groundwater level and slope displacement during typhoon rainfall. Slope stability was also analyzed during the period.

This research focuses on the dip slope area at the Huafan University in northeastern Taiwan. The campus is located on a geological dip slope toward the southwest in the Ta-Lun Shan area with an elevation range of 450 m to 550 m, as shown in Figure 2. For risk management and research on slope stability, monitoring systems were set up in 2001, which has collected geographical data since then. The monitoring systems

included inclinometers, tiltmeters, crack gauges, groundwater level observation wells, settlement and displacement monitoring marks, rebar strain gauges, concrete strain gauges, and rain gauges.

Jeng and Sue (2016) analyzed the monitoring data collected from more than 300 settlement and displacement observation marks on this site, and compared them with the displacement recorded by the inclinometers, thereby finding a preliminary relationship between the displacement of the slope and daily rainfall at the campus. The software STABL based on the limit equilibrium method was applied for analysis, concluding that a rise in the groundwater level caused by typhoons is the most critical factor in slope stability. Therefore, several countermeasures, including catchpits with horizontal drainage pipes, were recommended, and threshold-value curves of the slope displacement based on rainfall intensity and cumulative rainfall were established. Those curves were derived from the rainfall records of numerous typhoon events over the past ten years, along with the corresponding slope displacement increment recorded by the ground surface marks.

Considering the project budget, two catchpits were implemented within the study area in the present stage. The study provides valuable monitoring information and experience of continuous observation over years. The study also offers a comprehensive survey of the slope behavior during rainfall exposure. It includes the highest groundwater level measured within the slope, the time lag of response of groundwater variations, the influence of rainfall types and the amount, the relationship curve of rainfall amount and groundwater level, and a review of the effectiveness after the two catchpits were constructed. The displacement curve measured by the inclinometers was used for comparing feedback, and numerical analysis and simulation were performed to validate the changes in the behavior and mechanism. Subsequently, according to the characteristics of the slope, including the allowable displacement alert and action values, the data analyses revealed a corresponding relationship rainfall curve. The results will help to assess possible changes in groundwater level within the slope, possible slope displacement, and safety and stability factors, from the estimated rainfall before a disaster occurs.